# KLF4 Suppresses the Progression of Hepatocellular Carcinoma by Reducing Tumor ATP Synthesis through Targeting the Mir-206/RICTOR Axis

**DOI:** 10.3390/ijms25137165

**Published:** 2024-06-28

**Authors:** Yongjin Wang, Dinglan Zuo, Zhenkun Huang, Yuxiong Qiu, Zongfeng Wu, Shaoru Liu, Yi Zeng, Zhiyu Qiu, Wei He, Binkui Li, Yunfei Yuan, Yi Niu, Jiliang Qiu

**Affiliations:** 1State Key Laboratory of Oncology in South China, Guangdong Provincial Clinical Research Center for Cancer, Sun Yat-Sen University Cancer Center, 651 Dongfeng Road East, Guangzhou 510060, China; exshow@126.com (Y.W.); zuodl@sysucc.org.cn (D.Z.); huangzk@sysucc.org.cn (Z.H.); qiuyx@sysucc.org.cn (Y.Q.); wuzf@sysucc.org.cn (Z.W.); liusr1@sysucc.org.cn (S.L.); cengyi1@sysucc.org.cn (Y.Z.); qiuzhy3@mail3.sysu.edu.cn (Z.Q.); hewei@sysucc.org.cn (W.H.); libk@sysucc.org.cn (B.L.); yuanyf@mail.sysu.edu.cn (Y.Y.); 2Department of Liver Surgery, Sun Yat-Sen University Cancer Center, 651 Dongfeng Road East, Guangzhou 510060, China

**Keywords:** KLF4, hepatocellular carcinoma, ATP, RICTOR, miR-206

## Abstract

To address the increased energy demand, tumor cells undergo metabolic reprogramming, including oxidative phosphorylation (OXPHOS) and aerobic glycolysis. This study investigates the role of Kruppel-like factor 4 (KLF4), a transcription factor, as a tumor suppressor in hepatocellular carcinoma (HCC) by regulating ATP synthesis. Immunohistochemistry was performed to assess KLF4 expression in HCC tissues. Functional assays, such as CCK-8, EdU, and colony formation, as well as in vivo assays, including subcutaneous tumor formation and liver orthotopic xenograft mouse models, were conducted to determine the impact of KLF4 on HCC proliferation. Luciferase reporter assay and chromatin immunoprecipitation assay were utilized to evaluate the interaction between KLF4, miR-206, and RICTOR. The findings reveal low KLF4 expression in HCC, which is associated with poor prognosis. Both in vitro and in vivo functional assays demonstrate that KLF4 inhibits HCC cell proliferation. Mechanistically, it was demonstrated that KLF4 reduces ATP synthesis in HCC by suppressing the expression of RICTOR, a core component of mTORC2. This suppression promotes glutaminolysis to replenish the TCA cycle and increase ATP levels, facilitated by the promotion of miR-206 transcription. In conclusion, this study enhances the understanding of KLF4’s role in HCC ATP synthesis and suggests that targeting the KLF4/miR-206/RICTOR axis could be a promising therapeutic approach for anti-HCC therapeutics.

## 1. Introduction

The hallmark feature of cancer is the increased energy demand, which plays a pivotal role in tumor growth and metastasis [1]. Tumor cells undergo metabolic reprogramming, utilizing both oxidative phosphorylation (OXPHOS) and aerobic glycolysis to generate high levels of energy [2]. Targeting abnormal metabolism has been considered an attractive therapeutic strategy for HCC. 

Krüppel-like Factor 4 (KLF4) is a member of the evolutionarily conserved family of zinc finger transcription factors [3]. It was first discovered by Garrett-Sinha and Shields in 1996 in the differentiated epithelium, colon, and small intestine of newborn mice [4]. KLF4 has been extensively studied for its diverse roles in cellular physiology and pathology, including proliferation, differentiation, apoptosis [5], inflammation, and tumorigenesis [6,7]. In the context of HCC, KLF4 has been identified as a tumor suppressor [8]. It regulates multiple signaling pathways, such as the KLF4-pcadherin-GSK-β and KLF4-CD9/CD81-JNK axis [9,10], and suppresses migration and invasion through the KLF4-VDR pathway [11].

Despite extensive research on KLF4’s functions in various cellular processes and its role in HCC, our understanding of its role in metabolism remains limited. The underlying molecular mechanisms by which KLF4 affects metabolism require further investigation to comprehensively understand its functions in this context.

The mammalian target of rapamycin (mTOR) is the catalytic subunit of two protein complexes, mTOR complex 1 (mTORC1) and mTOR complex 2 (mTORC2) [12,13]. These complexes play crucial roles in regulating cell growth, proliferation, and metabolism [14]. RICTOR, a core component of mTORC2, functions as a hub at the mitochondria-associated endoplasmic reticulum (ER) membranes (MAM), regulating mitochondrial function and metabolism [15,16]. Deficiency of mTORC2 disrupts MAM, leading to mitochondrial defects such as increased mitochondrial membrane potential, ATP production, and calcium uptake [17]. In a study by Moloughney et al. (2016), it was revealed that mTORC2 responds to decreasing levels of glucose and glutamine catabolites by promoting glutaminolysis and preserving the tricarboxylic acid (TCA) cycle and hexosamine biosynthesis [18]. However, it remains unclear whether RICTOR is involved in the KLF4-mediated metabolic reprogramming in HCC.

MicroRNAs (miRNAs), small non-coding RNAs approximately 20–25 nucleotides in length, play crucial roles by directly binding to the 3′ untranslated regions (UTRs) of target mRNAs [19]. These molecules have gained significant attention due to their involvement in various biological processes and diseases, including cancer [20]. One specific miRNA, miR-206, has been studied for its dual roles as an oncogene or a tumor suppressor in different conditions [21]. Previous studies have shown that miR-206 exhibits anti-HCC effects by inhibiting the Cyclin-Dependent Kinase 9 (CDK9) signaling pathway [22]. Additionally, miR-206 acts as a sponge for Linc1749808, impacting Yes-associated protein 1 (YAP1) and epithelial-mesenchymal transition (EMT) in HCC cells [23]. The precise mechanism by which miR-206 regulates tumor energy metabolism in HCC remains unknown.

In this study, we demonstrate that KLF4 functions as a tumor suppressor in HCC. It is downregulated in tumor tissues, and low expression correlates with poor overall survival (OS) and recurrence-free survival (RFS) in HCC patients. Overexpression of KLF4 suppresses HCC cell proliferation both in vitro and in vivo. Interestingly, we find that KLF4 inhibits ATP synthesis. Specifically, KLF4 promotes the transcription of miR-206, which subsequently suppresses the expression of RICTOR. This suppression leads to a decrease in ATP synthesis, highlighting the novel role of KLF4 in regulating energy metabolism. Our results provide valuable insights into the molecular mechanisms of the KLF4/miR-206/RICTOR axis in cellular energetics. Furthermore, our findings suggest that. KLF4 may serve as a potential biomarker and therapeutic target for HCC.

## 2. Results

### 2.1. KLF4 Was Downregulated in HCC, and a Low Level of KLF4 Was Associated with a Poor Prognosis

To investigate the expression pattern and prognostic significance of KLF4 in HCC, we conducted a comprehensive analysis of mRNA and protein levels in both fresh and formalin-fixed paraffin-embedded (FFPE) tissue samples. The study cohort consisted of 14 pairs of fresh HCC tissues and adjacent non-tumor tissues (NTs), as well as 130 consecutive HCC tissue samples obtained from curative liver resection.

Firstly, using quantitative real-time polymerase chain reaction (qRT-PCR) and western blot analysis, we observed a significant downregulation of KLF4 mRNA and protein levels in HCC tissues compared to their matched NTs (Figure 1A,B). This finding was further validated in a larger cohort of 130 HCC tissue samples analyzed by immunohistochemistry (IHC). Consistent with our initial findings, KLF4 expression was significantly reduced in HCC tissues compared to paired adjacent liver tissues (*p* < 0.001; Figure 1C,D). Next, we investigated the association between KLF4 level and clinicopathological characteristics. We found that patients with high expression of KLF4 had a better tumor differentiation status (*p* = 0.008) and a lower incidence of microvascular invasion (MVI) (*p* = 0.033) (Appendix A). In univariate analyses, albumin < 35 g/L was identified as a risk factor for overall survival (hazard ratio [HR] = 4.69; 95% CI, 2.18–10.08, *p* < 0.001) and recurrence-free survival (hazard ratio [HR] = 3.02; 95% CI, 1.37–6.69, *p* = 0.006) (Appendix A). To investigate the expression of KLF4 in HCC, we utilized the TCGA and GSE14520 databases. We found that the mRNA level of KLF4 was significantly decreased in HCC tissues compared to paired adjacent liver tissues. (Figure 1E,F).

To assess the prognostic value of KLF4 expression in HCC, we performed Kaplan–Meier survival analyses. Lower KLF4 expression was associated with shorter overall survival (OS) (hazard ratio [HR] = 0.35; 95% confidence interval [CI] = 0.20–0.62, *p* < 0.001, Figure 1G) and recurrence-free survival (RFS) (HR = 0.33; 95% CI = 0.19–0.56, *p* < 0.001, Figure 1H) in our cohort. Additionally, analysis of the publicly available TCGA cohort confirmed our findings, demonstrating a positive association between KLF4 expression and RFS in HCC patients *(p =* 0.049, Figure 1I).

In summary, our results demonstrate that KLF4 expression is downregulated in HCC tissues and serves as an independent positive predictive indicator for both overall survival (OS) and recurrence-free survival (RFS).

### 2.2. KLF4 Suppressed HCC Growth In Vitro and In Vivo

To assess the biological functions of KLF4 in hepatocellular carcinoma (HCC), we initially examined the expression levels of KLF4 in HCC cell lines. Our findings revealed that MHCC-97H and MHCC-97L cells exhibited relatively higher expression of KLF4, while Hep3B and Huh7 cells displayed lower levels of KLF4 expression (Figure 2A). For further investigation, we used plasmids to stably overexpress KLF4 in Hep3B and Huh7 cells and employed two shRNAs to knock down KLF4 expression in MHCC-97H and MHCC-97L cells (Figure 2B).

Subsequently, we conducted EdU cell proliferation and clone formation assays to investigate the functional impact of KLF4 on HCC cells. In the EdU assays, we observed a significant increase in proliferating cells in the KLF4 knockdown cells compared to the controls, while the opposite effect was noted in the overexpression experiments (Figure 2C). These findings were further supported by CCK8 and colony formation assays, which demonstrated that KLF4 overexpression significantly suppressed HCC cell growth, while knockdown of KLF4 promoted HCC cell growth (Figure 2D,E, respectively).

To further assess the influence of KLF4 on HCC progression in vivo, we utilized Hep3B cells overexpressing KLF4 and MHCC-97H cells with inhibited KLF4 expression in mouse models (Figure 2F). In a subcutaneous transplantation HCC mouse model, the tumors in the KLF4 overexpression group exhibited significantly smaller sizes and lower weights compared to those in the vector group (Figure 2G). Conversely, knockdown of KLF4 resulted in increased tumor sizes and weights compared to the control group (Figure 2H). Comparable results were also obtained in an orthotopic liver tumor xenograft mouse model (Figure 2I). Collectively, these results indicate that KLF4 suppresses HCC growth (Figure 2J,K).

### 2.3. KLF4 Reduced ATP Synthesis in HCC Cells

To elucidate the potential molecular mechanism underlying KLF4-mediated HCC suppression, we conducted a functional enrichment analysis of the TCGA dataset. The results revealed that KLF4 was involved in the regulation of multiple pathways in HCC cells, particularly ATP synthesis (Figure 3A). Further Gene Set Enrichment Analysis (GSEA) of the TCGA dataset showed significant enrichment of ATP synthesis in HCC patients with low KLF4 expression (*p* < 0.001, normalized enrichment score (NES) = 2.41; Figure 3B). To confirm these findings, we measured ATP concentration and found that overexpressing KLF4 in Hep3B and Huh7 cells led to a reduction in ATP production, while knockdown of KLF4 resulted in an upregulation of ATP production in MHCC-97H and MHCC-97L cells (Figure 3C). 

Cancer cells typically generate ATP through both aerobic glycolysis and oxidative phosphorylation (OXPHOS) to support cellular activities. To assess the impact of KLF4 on glucose metabolism in HCC cells, we utilized the Seahorse extracellular flux assay to measure glucose-induced extracellular acidification rate (ECAR), a proxy for glycolytic rate and capacity. Overexpression of KLF4 in Hep3B and Huh7 cells resulted in a significant decrease in glycolysis, glycolytic capacity, and glycolytic reserve. Conversely, the knockdown of KLF4 in MHCC-97H and MHCC-97L cells led to an increase in both glycolysis and glycolytic capacity (Figure 3D). Additionally, to investigate the effect of KLF4 on mitochondrial respiration in HCC cells, we measured glucose-induced O2 consumption rate (OCR) using the Seahorse extracellular flux assay. Overexpression of KLF4 in Hep3B and Huh7 cells resulted in a marked decrease in basal OCR, ATP-linked OCR, and maximal OCR. Conversely, the knockdown of KLF4 in MHCC-97H and MHCC-97L cells led to an increase in all these parameters (Figure 3E). These findings suggest that KLF4 inhibits the progression of HCC cells through the regulation of ATP synthesis.

### 2.4. KLF4 Suppressed the Progression of HCC Dependent on RICTOR 

To further investigate the molecular mechanisms by which KLF4 regulates ATP synthesis in HCC, we conducted untargeted metabolome profiling of MHCC-97H cells with either KLF4 knockdown or control conditions, followed by a KEGG pathway analysis. This analysis revealed a significant enrichment of the mTOR signaling pathway (Figure 4A). The mTOR signaling pathway is aberrantly activated in tumors and controls cancer cell metabolism by altering the activity of various key metabolic enzymes. The mTORC1 complex consists of mTOR, Raptor, mLST8, PRAS40, and DEPTOR, while the mTORC2 complex consists of mTOR, Rictor, DEPTOR, mSin1, and Protor1/2 [24]. To validate the impact of KLF4 on the mRNA expression of genes involved in the mTOR complex, qRT-PCR assays were conducted. The results demonstrated that the inhibition of KLF4 upregulated the mRNA level of RICTOR in both MHCC-97H and MHCC-97L cells (Figure 4B). Western blot analysis further confirmed that the knockdown of KLF4 in MHCC-97H and MHCC-97L cells resulted in an upregulation of RICTOR expression (Figure 4C). RICTOR, as a core component of mTORC2, promotes glutaminolysis to refill the TCA cycle, thereby increasing ATP levels [25].

To assess the prognostic value of RICTOR expression in HCC, we conducted Kaplan–Meier survival analyses utilizing the publicly available TCGA cohort. We observed that higher RICTOR expression was associated with shorter overall survival (OS) (hazard ratio [HR] = 1.756; 95% confidence interval [CI] = 1.229–2.510, *p* = 0.002, Figure 4D) and recurrence-free survival (RFS) (HR = 1.54; 95% CI = 1.147–2.069, *p* = 0.00412, Figure 4E).

Hence, we investigated whether RICTOR was responsible for the KLF4-induced suppression of HCC. We targeted RICTOR with siRNA (siRICTOR) in KLF4-silenced MHCC-97H cells. The subsequent results from EdU staining, CCK-8 assay, and colony formation assays indicated that RICTOR depletion significantly abrogated the increased cell proliferation caused by the knockdown of KLF4 (Figure 4F–H). Additionally, in vivo experiments further confirmed that RICTOR knockdown effectively negated the impact of KLF4 silencing, as evident from weekly tumor volume measurements and final tumor weight determinations (Figure 4I). Collectively, these findings suggest that KLF4 suppresses the progression of HCC dependent on RICTOR.

### 2.5. KLF4 Regulated RICTOR Expression through miR-206 in HCC

As a transcription factor (TF), KLF4 primarily functions in promoting gene transcription. However, our research has revealed a contrasting role for KLF4, wherein it inhibits the expression of RICTOR. This intriguing finding indicates an alternative mechanism governing the interaction between KLF4 and RICTOR beyond transcriptional regulation. MiRNAs, which exert inhibitory effects by directly binding to the untranslated regions (UTRs) of target mRNAs, provide a potential avenue for the interaction between KLF4 and RICTOR. Therefore, we conducted a bioinformatics analysis to further investigate this possibility. 

By conducting rigorous bioinformatics analysis using databases such as miRWalk (http://mirwalk.umm.uni-heidelberg.de/ (accessed on 2 November 2023)) and TargetScan (http://www.targetscan.org (accessed on 2 November 2023)), we identified several microRNAs, namely hsa-mir-206, hsa-mir-143, hsa-mir-15a, hsa-mir-182, hsa-mir-183, and hsa-let-7i-5p, that may play a crucial role in the linkage between KLF4 and RICTOR (Figure 5A). The qRT-PCR experiments further corroborated this finding, revealing a decrease in mRNA expression of miR-206-associated genes upon KLF4 knockdown (Figure 5B). Using TargetScan, we identified a conserved binding site for miR-206 in the 3′UTR region of the RICTOR gene (Figure 5C). To investigate the direct regulatory relationship between RICTOR and miR-206, we constructed a luciferase reporter plasmid harboring the 3′UTR of RICTOR (Figure 5D). As shown in Figure 5D, luciferase activity was significantly reduced in the reporter containing the wild-type 3′UTR upon co-transfection with miR-206 mimics. Conversely, no significant difference was observed in luciferase activity between the mutant-type 3′UTR and the control mimics. Western blot analysis (Figure 5E) further confirmed a significant decrease in RICTOR protein expression upon miR-206 overexpression, indicating that RICTOR was indeed directly regulated by miR-206.

Furthermore, we conducted an analysis of the miR-206 upstream region (−1 to−2000 kb) using JASPAR (http://jaspar.genereg.net/ (accessed on 24 November 2023)) and predicted the presence of five KLF4 binding sites in the putative promoter region (Figure 5F). To validate this prediction, we performed a chromatin immunoprecipitation (ChIP) assay using an anti-KLF4 antibody. The results demonstrated a direct interaction between KLF4 and the miR-206 promoter (Figure 5G). Notably, nucleotides (nts) −1616/−1606 of the miR-206 promoter region were identified as a putative binding site for KLF4 (KLF4 site 2). Taken together, these findings provide strong evidence that KLF4 transcriptionally regulates miR-206 expression in HCC cell.

In conclusion, our study demonstrates that KLF4 functions as a tumor suppressor and inhibits the progression of HCC through the miR206/RICTOR axis. These findings provide a potential therapeutic approach for HCC (Figure 6).

## 3. Materials and Methods 

### 3.1. Human Tissue Specimens 

A total of 130 patients with HCC who underwent curative surgery at the Sun Yat-sen Cancer Center (SYSUCC; Guangzhou, China) between January 2010 and June 2012 were enrolled in the study. None of the patients received any local or systemic anticancer therapies before surgical resection, and no postoperative anticancer treatments were administered before any identified relapse. The diagnosis of all cases was confirmed pathologically based on the terminology criteria established by the International Working Party. Matched tumor and adjacent normal tissue samples were collected from these 130 enrolled patients for the construction of the tissue microarray. Overall survival (OS) was defined as the time interval between the last followup or death and surgery, while disease-free survival (DFS) was defined as the time interval between surgery and tumor recurrence or last follow-up. All samples were obtained with the informed consent of the patients.

### 3.2. Immunohistochemistry (IHC)

HCC specimens were selected and re-embedded into new paraffin blocks for tissue microarray (TMA) construction. The TMA blocks were cut into 4 μm sections and underwent IHC staining. The rabbit polyclonal antibody used was anti-KLF4 (1:100; Hpa002926, MERCK, St. Louis, MO, USA). Based on IHC, positive staining was quantified and classified into four categories: ≤25% for 1; 26 to 50% for 2; 51 to 75% for 3; and ≥76% for 4. Staining intensity was graded as negative (scored as 0), weak (1), moderate (2), or strong (3). Two pathologists independently reviewed all scores, and expression levels were defined by the sum of the grades for the percentage of positive staining and intensity. The median of the IHC score was chosen as the cut-off value for the high (>6) and low (≤6) KLF4 groups.

### 3.3. Cell Culture and Treatments

HCC cell lines (Huh7, Hep3B, MHCC-97H, and MHCC-97L) and human embryonic kidney cells (HEK293T) were purchased from Guangzhou Jenniobio Biotechnology (Guangzhou, China) with STR (short tandem repeat) appraisal certificates. Cells were maintained in Dulbecco’s Modified Eagle medium (DMEM; Thermo Fisher, Grand Island, NY, USA) supplemented with 10% fetal bovine serum (FBS; Gibco, Auckland, New Zealand) at 37 °C in 5% CO_2_.

### 3.4. Cell Proliferation Assays

Three experiments were performed to analyze cell proliferation ability. In the Cell Counting Kit-8 (CCK-8; DojinDo, Kumamoto, Japan) assay, 1000 cells were seeded in 96-well plates with six repeated wells for each experimental condition. Two hours after replacing the supernatant with fresh medium containing CCK-8 reagent in a 10:1 ratio, the absorbance was measured by the Biotek Epoch 2 machine (BioTek, Winooski, VT, USA) at 450 nm.

EdU cell proliferation staining was performed using an EdU kit (C0071S; Beyotime Biotechnology, China). Briefly, indicated cells were incubated with the EdU buffer for 3 h, fixed by 4% polyformaldehyde, and stained the nucleus with Hoechst. The results were observed and captured using a fluorescence microscope (Olympus Corporation, Tokyo, Japan). Cell proliferation staining was performed using an EdU kit (C0071S; Beyotime Biotechnology, Shanghai, China). Briefly, the indicated cells were incubated with the EdU buffer for 3 h, fixed with 4% paraformaldehyde, and then the nuclei were stained with Hoechst. The results were observed and captured using a fluorescence microscope (Olympus Corporation, Tokyo, Japan).

For the colony formation assay, 500–800 indicated stable cells were plated per well in 6-well plates. Fourteen days later, the cells were fixed in methanol and stained with 0.1% crystal violet. Colony images were photographed and counted for contrast. 

### 3.5. Animal Experiments

All animal research procedures were performed in accordance with the detailed rules of the Animal Care and Use Ethics Committee of SYSUCC, with efforts made to minimize animal suffering. Male nude mice (Guangdong Medical Lab Animal Center) were purchased at 3–4 weeks of age and kept under specific pathogen-free (SPF) conditions.

To evaluate the inhibitory effect of KLF4 on tumor growth in vivo, mice were randomly divided into designated groups (*n* = 5 per group) before injection without the use of a blinding method. Hep3B and MHCC-97H stable cells were then subcutaneously injected into the right side of the posterior flank of the same male BALB/c athymic nude mice (1.0 × 10^7^ cells/mouse) to establish the subcutaneous model. Body weight and tumor size were monitored every three days. Eleven or thirteen days after inoculation, all mice (*n* = 5 per group) were euthanized for further analysis. Tumor volume (V) was measured using the formula: V = (L × W2)/2 (where L = length; W = width).

For the liver orthotopic xenograft mouse model, 1 × 10^6^ shCtrl, shKLF4#1, or shKLF4#2 MHCC-97H cells, and control or KLF4-OE stable Hep3B cells were resuspended in 100% Matrigel (BD Biocoat, Corning, NY, USA) and injected into the left lobes of the livers in 5- to 6-week-old male NOD/SCID mice. The incision was then closed using surgical suture threads with a needle and medical adhesive bandage. At six weeks post-inoculation, tumor-bearing mice were sacrificed, and livers were harvested for histological analysis.

### 3.6. Seahorse

The mitochondrial oxygen consumption rate (OCR) and extracellular acidification rate (ECAR) were determined using an XF-24 Extracellular Flux Analyzer (Seahorse Bioscience, Agilent Technologies, Santa Clara, CA, USA). Cells were seeded in a 24-well Seahorse XF Cell Culture Microplate (200,000 cells per well). After 24 h, the DMEM was changed to a pre-warmed assay medium, and the cell culture microplate was placed in a 37 °C non-CO2 incubator for 45 min. The OCR was measured in the XF Analyzer according to the manual of the Seahorse XF Cell Mito Stress Test Assay (Seahorse Bioscience, Agilent Technologies, Santa Clara, CA, USA). The ECAR was measured in the XF Analyzer according to the manual of the Seahorse XF Cell Glycolysis Stress Test Assay (Seahorse Bioscience, Agilent Technologies, Santa Clara, CA, USA).

### 3.7. Chromatin Immunoprecipitation (ChIP) Assay

ChIP assay was performed using the Simple ChIP Enzymatic Chromatin IP kit (Cell Signaling Technology, Boston, MA, USA). HCC cells were crosslinked with formaldehyde, lysed with SDS buffer, and then subjected to ultrasonication. Subsequently, the cells were incubated with specific antibodies (1:100; #12173, Cell Signaling Technology, Boston, MA, USA) or normal rabbit IgG. After washing with high salt and low salt, the DNA was eluted and de-crosslinked, and enrichment was examined using qRTPCR.

For the ChIP assay, a ChIP kit (Cell Signaling Technology, Boston, MA, USA) was used according to the manufacturer’s instructions. Samples were crosslinked with 1% formaldehyde (Sigma-Aldrich, St. Louis, MO, USA) solution, followed by quenching the reaction with glycine solution. The cells were then lysed, and the nucleoprotein complexes were sonicated for 10 cycles of 10 s power-on and 20 s intervals with an intensity of 200 W using the sonicate conductor (Qsonica, Newtown, CT, USA). Subsequently, an anti-KLF4 antibody or IgG was added and incubated with the complexes overnight at 4 °C. The next day, Protein A/G magnetic beads were added to precipitate the indicated fragments for 4 h at 4 °C. After extraction and purification of the DNA, qRT-PCR was performed to identify the region interacting with the miR-206-specific primers. The primers used are listed in Appendix A. The experiments were performed in triplicate, and the amount of immunoprecipitated DNA was normalized to the input.

### 3.8. Luciferase Reporter Assay

To investigate the effect of miR-206 on the Rictor 3′-UTR, HEK293T cells in a 6-well plate were co-transfected with 10 nM of miR-206 or NC duplex, 1 µg pEZX-RICTOR-3′-UTR-WT or pEZX-RICTOR-3′-UTR-MUT and cultured for 24 h. The cells were then trypsinized and transferred to a 96-well plate. After 24 h, a luciferase reporter assay was performed following the manufacturer’s instructions. Renilla luciferase activity was normalized to firefly luciferase activity. Cell lysates were collected 48 h after transfection and subjected to luciferase activity assays using the luciferase reporter system (Genecopoeia, Guangzhou, China). The data are presented as the mean ± standard error of the mean (SEM) from at least three independent experiments.

### 3.9. Quantitative RT Real-Time PCR (qRT-PCR)

Total RNA was extracted from the HCC tissues and cell lines using the Trizol reagent kit (Life Technologies, Carlsbad, CA, USA). After treatment with DNase I (TaKaRa, Dalian, China), 2 µg of total RNA was used for cDNA synthesis with random hexamers and Superscript III (Invitrogen, Grand Island, NY, USA). The cDNA templates were then subjected to qRT-PCR amplification using the SYBR Green qRT-PCR kit (Invitrogen, Carlsbad, CA, USA). The sequence of the primers used can be found in Appendix A.

The qRT-PCR analysis of miR-206 was performed on an ABI PRISM 7900 Sequence Detector using the SYBR Green qRT-PCR Kit (Applied Biosystems, Carlsbad, CA, USA). The following primers were used for detection: HmiRQP0159 for miR-130b-3p and HmiRQP9001 for U6 (Genecopoeia, Guangzhou, China). All reactions were run in triplicate, and the cycle threshold (Ct) values should not differ by more than 0.5 among triplicates. The miR-206 level was normalized to RNU6B, which yielded a 2-ΔΔCt value. 

### 3.10. Immunoblotting (IB)

Cells and tissues were lysed with RIPA lysis buffer. Proteins were extracted, loaded onto SDS-PAGE, and transferred onto the PVDF membrane (Millipore, Billerica, MA, USA). After blocking with 5% skimmed milk (Beyotime, Shanghai, China) and sequential incubation with primary and secondary antibodies (anti-RICTOR, 6867-2-IG, 1:1000; anti-GAPDH, 60004-1-Ig, 1:2000; Proteintech, Wuhan, China); (anti-KLF4, 12173S, 1:1000; anti-rabbit IgG, 7074S, 1:3000; anti-mouse IgG, 7076S, 1:3000. Cell Signaling Technology, Boston, MA, USA) the blots were detected using the ECL detection kit (Millipore; Boston, MA, USA).

### 3.11. Plasmids for Overexpression and Knockdown

Plasmids for expressing or knocking down human KLF4 were purchased from Kidan Biosciences Co., Ltd. (Guangzhou, China). Transfection was performed using Lipofectamine 2000 (Invitrogen) according to the manufacturer’s instructions. Briefly, cells were seeded at 1.0 × 10^6^ cells per well in a 6-well tissue culture dish or at 3 × 10^6^ cells per 10 cm tissue culture dish and transfected with 2 μg or 12 μg plasmid DNA for 24 h, respectively. The sequence of shKLF4-1#: CCGGCTGACCAGGCACTACCGTAAACTCGAGTTTACGGTAGTGCCTGGTCAGTTTTTG. The sequence of shKLF4-2#: CCGGCGCCACCCACACTTGTGATTACTCGAGTAATCACAAGTGTGGGTGGCGTTTTTG.

### 3.12. Small Interfering RNA

Small interfering RNA for RICTOR was purchased from Genepharma (Shanghai, China). Transfection of small interfering RNA was performed with Lipofectamine-RNAiMAX (Invitrogen, Carlsbad, CA, USA). After 6-8 h, the supernatant was replaced with fresh medium, and the efficiency was identified by qRT-PCR and western blotting 72 h after transfection. The sequence of siRICTOR is CGAGGACCUAAGCCUUAUATT, with its anti-sense sequence being UAUAAGGCUUAGGUCCUCGTT.

### 3.13. Data Availability

The human cancer KLF4 and RICTOR expression data were derived from the TCGA Research Network (http://cancergenome.nih.gov/ (accessed on 21 October 2023)). The bulk RNA-seq data supporting the findings of this study have been deposited in the Gene Expression Omnibus under accession numbers GSE14520.

### 3.14. Statistical Analysis

The experiments were repeated at least three times independently, and the measured data were represented as the mean ± SD. Binary variables were compared using the Chi-squared test. The Student’s *t*-test or the Mann-Whitney U test was used to compare values between subgroups. Survival curves were constructed using the Kaplan-Meier method and analyzed by the log-rank test. Significant prognostic factors identified by univariate analysis were entered into a multivariate analysis using the Cox proportional hazards model. All analyses were two-sided, and *p*-values of less than 0.05 were considered significant. Statistical analyses were performed using the Statistical Package for Social Sciences version 25 (SPSS Inc., Chicago, IL, USA) and GraphPad Prism 7.0 software (GraphPad, Inc., La Jolla, CA, USA).

## 4. Discussion

As tumor cells continuously proliferate, they face the challenge of meeting their increasing energy demands [1]. Understanding the fundamental mechanism of energy metabolism in HCC is crucial for developing new therapeutic strategies. In our study, we discovered that KLF4 functions as a novel metabolic regulator that is commonly downregulated in HCC tissues. The reduced expression of KLF4 is associated with overall survival (OS) and recurrence-free survival (RFS) in HCC patients. Moreover, the downregulation of KLF4 expression promotes ATP synthesis and enhances the growth of HCC cells both in vitro and in vivo. Mechanistically, KLF4 suppresses HCC growth by regulating RICTOR, which is modulated by the KLF4/miR-206/mTOR axis, thereby inhibiting ATP synthesis in HCC.

KLF4, a member of the evolutionarily conserved family of zinc finger transcription factors, has been extensively studied for its regulatory roles in various cellular physiological processes, including proliferation, differentiation, and apoptosis [5,26]. Furthermore, KLF4 has been implicated in inflammation and tumorigenesis [6,7] and has been identified as a tumor suppressor in HCC [8]. Mechanistically, KLF4 has been shown to inhibit HCC progression through the regulation of signaling pathways such as KLF4-pcadherin-GSK-3β and KLF4-CD9/CD81-JNK [9,10]. Additionally, KLF4 inhibits the migration and invasion of tumor cells by suppressing biological enzymes like TIMP-1 and TIMP-2 [27]. Despite being characterized as a tumor suppressor gene in HCC, the biological function of KLF4 in metabolism remains poorly understood. Moreover, the underlying molecular mechanism remains largely unknown. In this study, we aimed to investigate the biological function and molecular mechanism of KLF4 in HCC metabolism. We demonstrated a significant inhibitory effect of KLF4 on HCC proliferation both in vitro and in vivo, highlighting its potential tumor-suppressive role in HCC development. Abnormal energy metabolism is a critical aspect of HCC tumorigenesis. Based on our research, we discovered that the downregulation of KLF4 promotes ATP synthesis. However, the signaling pathways and target genes involved in the inhibition of ATP synthesis by KLF4 are still unclear. To explore the potential molecular mechanism, we conducted untargeted metabolome profiling, which revealed the involvement of KLF4 in the mTOR signaling pathway. We confirmed RICTOR as a target gene of KLF4. RICTOR, a component of mTORC2, plays a crucial role in cell proliferation and metabolism [15,16], particularly in the tricarboxylic acid (TCA) cycle and hexosamine biosynthesis [24]. In this study, we demonstrated that KLF4-mediated downregulation of RICTOR suppresses cell proliferation in HCC cells, involving the regulation of the mTORC2 signaling pathway.

Transcription factors typically promote gene/miRNA transcription, and it has been reported that RICTOR is regulated at the transcriptional level by miRNA in various tumors. Through in silico target prediction algorithms, a complementary binding sequence for miR-206 was identified in the 3′-UTR of RICTOR mRNA. Previous studies have demonstrated that miR-206 is downregulated in liver CSCs and exerts its inhibitory effect on CSC growth by directly targeting the Epidermal Growth Factor Receptor (EGFR) [28]. MiR-206 also exerts an anti-HCC effect by inhibiting the CDK9 signaling pathway, suggesting that the miR206-CDK9 pathway could be a potential therapeutic target for HCC [22]. In this study, we provide evidence that KLF4 regulates HCC progression through the transcriptional activation of miR-206. Knockdown of KLF4 resulted in a significant decrease in cellular miR-206 expression. Additionally, ChIP assays demonstrated the interaction between KLF4 and the miR-206 promoter sequence. Our study revealed that miR-206 directly targets RICTOR and inhibits ATP synthesis in HCC. Taking into account previous findings from other research groups and the results of our study, it can be concluded that KLF4 is regulated by the miR206/RICTOR axis, leading to the inhibition of ATP synthesis in HCC through the mTOR signaling pathway.

The mTOR pathway is commonly activated in human cancers and plays a crucial role in tumor progression, emphasizing the need for further investigation into the molecular mechanism of abnormal metabolism in HCC. While our study has revealed that down-regulation of KLF4 activates the mTOR pathway, the specific mechanism by which KLF4 regulates this pathway requires further elucidation.

In summary, this study provides mechanistic evidence supporting the role of KLF4 in regulating RICTOR expression through miR-206. Moreover, the KLF4/miR-206/RICTOR axis is shown to have a critical function in ATP synthesis in HCC cells. Our study not only identifies a novel molecular mechanism underlying ATP synthesis but also highlights the aberrant KLF4/miR-206/RICTOR signaling pathway as a promising molecular target for the development of novel therapeutic strategies to control the development and progression of HCC.

## Figures and Tables

**Figure 1 ijms-25-07165-f001:**
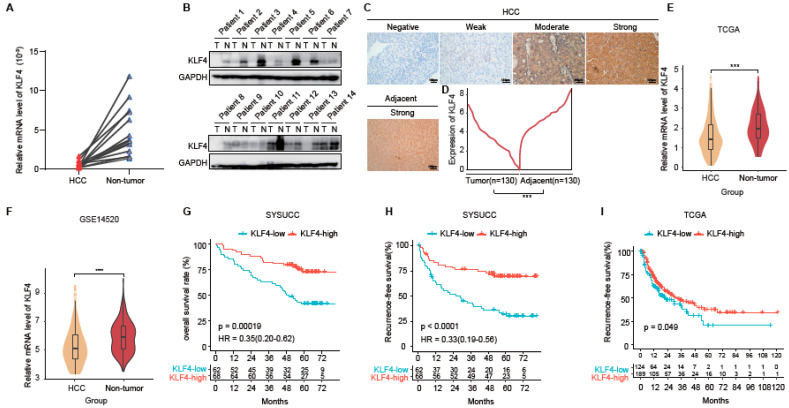
KLF4 was downregulated in HCC, and a low level of KLF4 was associated with poor prognosis. (**A**) The qRT-PCR and (**B**) IB were used to analyze the expression of KLF4 in 14 paired hepatocellular carcinoma (HCC) and adjacent non-tumor tissues. (**C**) Representative images of KLF4 expression in hepatocellular carcinoma (HCC) and adjacent normal liver tissues are shown. The scale bar indicates 100 µm. (**D**) KLF4 expression was downregulated in HCC tissues compared to adjacent non-tumor liver tissues (n = 130, paired *t*-test). (**E**,**F**) Relative mRNA levels of KLF4 in HCC and normal tissues were determined based on the TCGA and GSE14520 datasets. (**G**,**H**) Kaplan–Meier analysis revealed that low KLF4 expression was associated with shorter overall survival (OS) and recurrence-free survival (RFS) curves in the SYSUCC HCC cohort (n = 130). (**I**) Kaplan–Meier analysis revealed that low KLF4 was associated with shorter recurrence-free survival (RFS) curves in the TCGA HCC cohort (n = 276). Statistical results are presented as mean ± SD (from triplicates), and significance was determined by Student’s *t*-test *(**** *p* < 0.001, and **** *p* < 0.0001).

**Figure 2 ijms-25-07165-f002:**
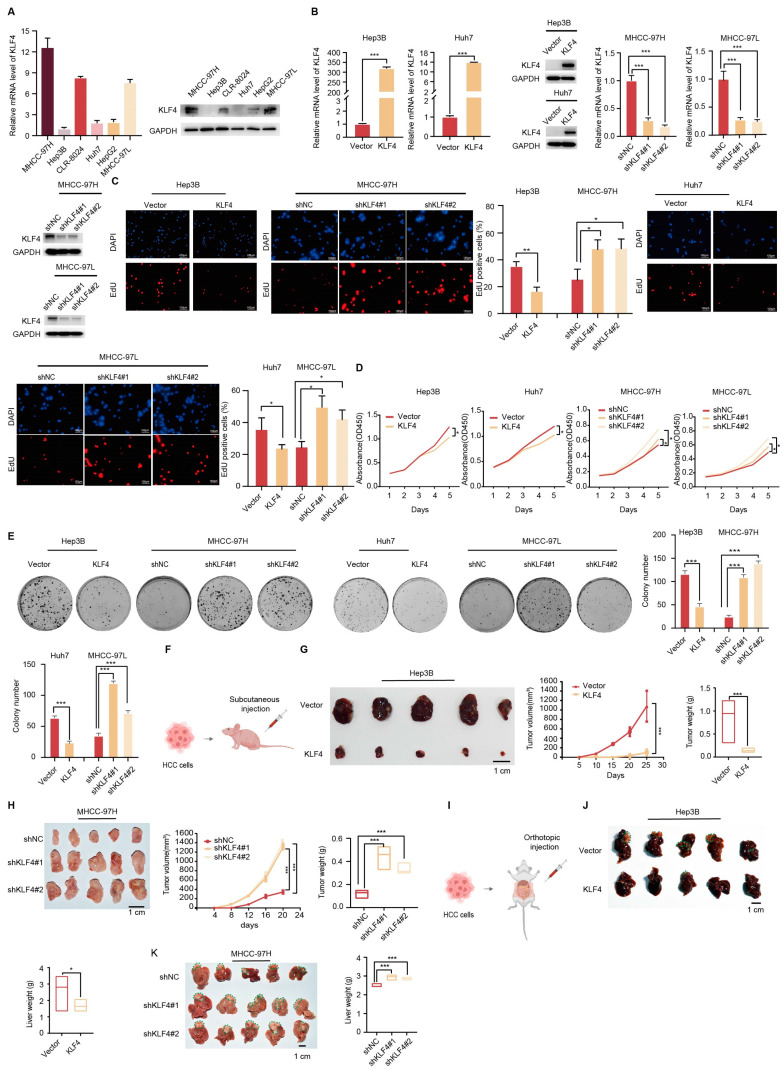
KLF4 suppressed HCC growth in vitro and in vivo. (**A**) qRT-PCR (top) and Western blot (bottom) were used to show the expression of KLF4 in 6 HCC cell lines. (**B**) The efficiency of KLF4 overexpression and knockdown was measured using qRT-PCR (top) and Western blotting (bottom) assays. (**C**–**E**) KLF4 suppressed the cell growth of HCC cells, as evaluated by EdU assay (**C**), CCK-8 assay (**D**), and colony formation assays (**E**). (**F**) Schematic diagram of the subcutaneous xenograft HCC mouse model. (**G**) KLF4 overexpression suppressed the proliferation of HCC cells in nude mice. Representative images, tumor volume, and tumor weight were assessed after the subcutaneous injection of Hep3B cells. Five mice were included in each group. (**H**) KLF4 knockdown promoted the proliferation of HCC cells in nude mice. Representative images, tumor volume, and tumor weight were assessed after the subcutaneous injection of MHCC-97H cells. Five mice were included in each group. (**I**) Schematic diagram of the orthotopic liver tumor xenograft mouse model. (**J**,**K**) KLF4 slightly suppressed liver tumor growth in the orthotopic xenograft mouse model, (**J**) Stable KLF4–overexpression Hep3B cells and control cells were injected into the livers of each NOD/SCID mouse, (**K**) Stable KLF4-knockdown MHCC-97H cells and control cells were injected into the livers of each NOD/SCID mouse. After six weeks of injection, the livers were dissected for pathological analysis. Tumors in the livers were indicated by green circles, and the liver weight was measured. Statistical results are presented as mean ± SD (from triplicates), and significance was determined by Student’s *t*-test *(** *p* < 0.05, ** *p* < 0.01, *** *p* < 0.001).

**Figure 3 ijms-25-07165-f003:**
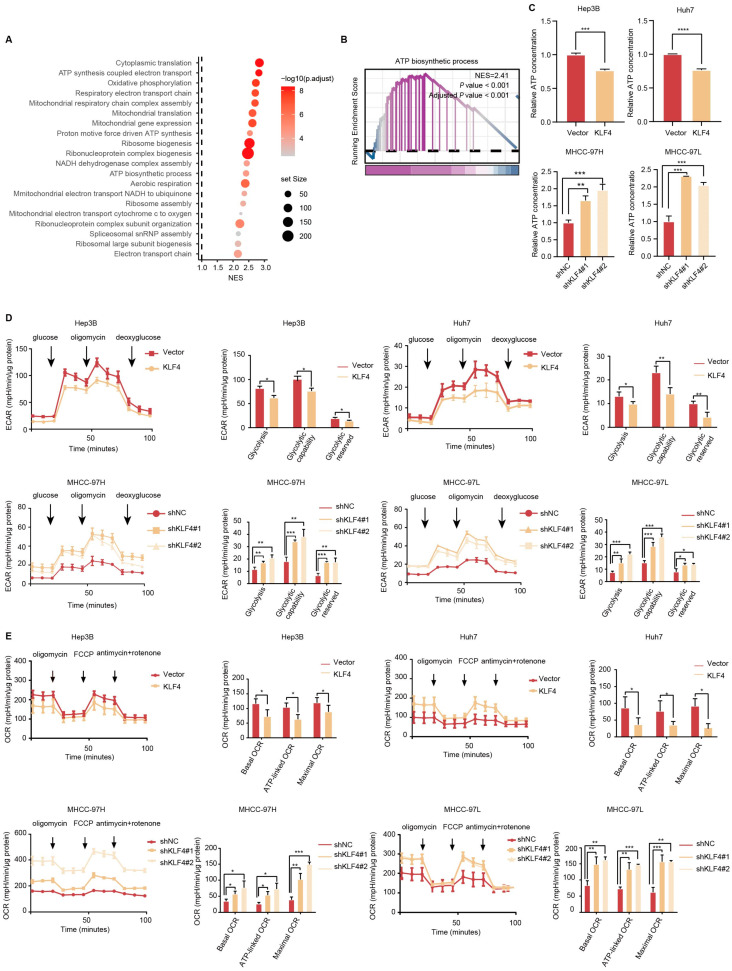
KLF4 reduced ATP synthesis in HCC cells. (**A**) KEGG pathway analysis revealed the significantly affected signaling pathways in the TCGA database. (**B**) Gene set enrichment analysis (GSEA) of transcriptome data in the TCGA database revealed a negative correlation between levels of KLF4 and ATP synthesis. The normalized enrichment score (NES) and false-discovery rate (FDR) were used for analysis. (**C**) KLF4 suppressed the ATP concentration in HCC cells, as evaluated by the relative ATP concentration in the indicated group. (**D**) Effects of KLF4 on glycolysis in HCC cells determined by Seahorse ECAR analyzer. The extracellular acidification rate over time, glycolysis, glycolytic capability, and glycolytic reserve were shown. (**E**) Effects of KLF4 on oxidative phosphorylation in HCC cells determined by Seahorse XF analyzer. The oxygen consumption rate over time, basal respiration rate, ATP-linked respiration rate, and maximal respiration rate were shown. Statistical results are presented as mean ± SD (from triplicates), and significance was determined by Student’s *t*-test *(** *p* < 0.05, ** *p* < 0.01 and *** *p* < 0.001, and **** *p* < 0.0001).

**Figure 4 ijms-25-07165-f004:**
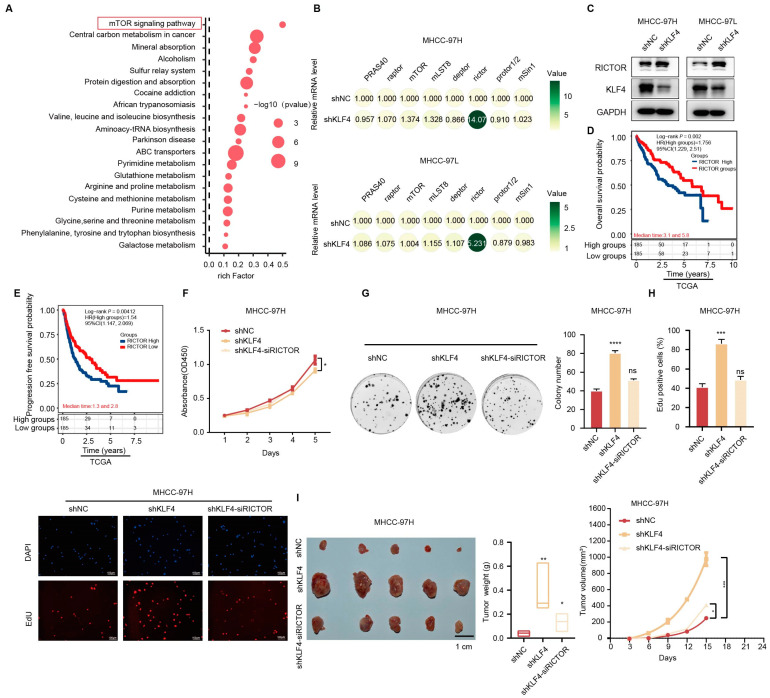
KLF4 suppresses the progression of HCC through a RICTOR way. (**A**) The down-regulation of KLF4 in HCC activated the mTOR signaling pathway. Top 20 statistics of KEGG pathway enrichment based on the differentially expressed genes (DEGs) after KLF4 knockdown in MHCC-97H cells. In the scatter plot, RichFactor was the ratio of DEG numbers noted in this pathway term to all gene numbers noted in this pathway term, indicating intensiveness. *p*-value ranges from 0 to 1, and a lower *p*-value represents greater intensity. The mTOR signaling pathway was one of the most regulated biofunctions upon KLF4 knockdown. (The red frame indicated the selected pathway in this study.) (**B**) The relative mRNA levels of genes in the mTOR complex were analyzed in MHCC-97H cells (**top**) and MHCC-97L cells (**bottom**). (**C**) Western blot assays showing the relative levels of KLF4 and RICTOR after KLF4 knockdown in MHCC-97H and MHCC-97L cells. (**D**,**E**) Kaplan-Meier analysis revealed that higher RICTOR expression was associated with shorter overall survival (OS) and recurrence-free survival (RFS) curves in the TCGA HCC cohort (n = 276). (**F**–**H**) Knockdown of RICTOR attenuated the KLF4-induced suppression of HCC cell growth, as evaluated by CCK8 assay (**F**), colony formation assays (**G**), and EdU assay (**H**). (**I**) Representative images, tumor volume, and tumor weight were assessed after the subcutaneous injection of MHCC-97H cells. Five mice were included in each group. Statistical results are presented as mean ± SD (from triplicates), and significance was determined by Student’s *t*-test *(** *p* < 0.05, ** *p* < 0.01, *** *p* < 0.001 and **** *p* < 0.0001, ns = no significance).

**Figure 5 ijms-25-07165-f005:**
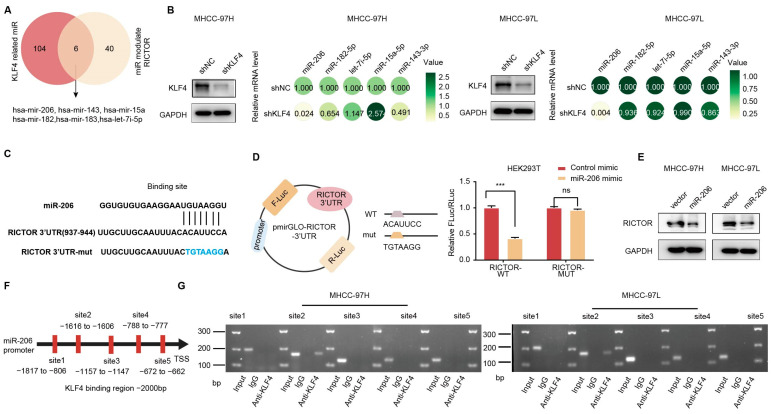
KLF4 regulated RICTOR expression through miR-206 in HCC, (**A**) Venn diagrams were used to show the overlapped essential miRNAs regulated by KLF4 and those that regulate RICTOR. (**B**) MiR-206 was identified as the link between KLF4 and RICTOR, as indicated by the relative mRNA levels in MHCC-97H and MHCC-97L cells. (**C**) Luciferase reporter plasmids were constructed containing the wild-type or mutant RICTOR 3′-UTR sequences immediately downstream of the luciferase gene. The sequences of the predicted miR-206 binding site within the RICTOR 3′-UTR, including wild-type and mutant sequences (underlined), are shown. (The alphabets in blue indicate the mutated site.) (**D**) Relative luciferase activity was analyzed in HEK293T cells transfected with the indicated reporter plasmids and either the miR-206 mimic or control. (**E**) Overexpression of miR-206 downregulated endogenous RICTOR protein level in MHCC-97H and MHCC-97L cells. (**F**) Schematic representation of the miR-206 promoter regions and the primers used for the ChIP assay were indicated. (**G**) ChIP samples were analyzed by qRT-PCR in MHCC-97H cells (**G**, **left**) and MHCC-97L cells (**G**, **right**). Statistical results are presented as mean ± SD (from triplicates), and significance was determined by Student’s *t*-test *(**** *p* < 0.001, ns = no significance).

**Figure 6 ijms-25-07165-f006:**
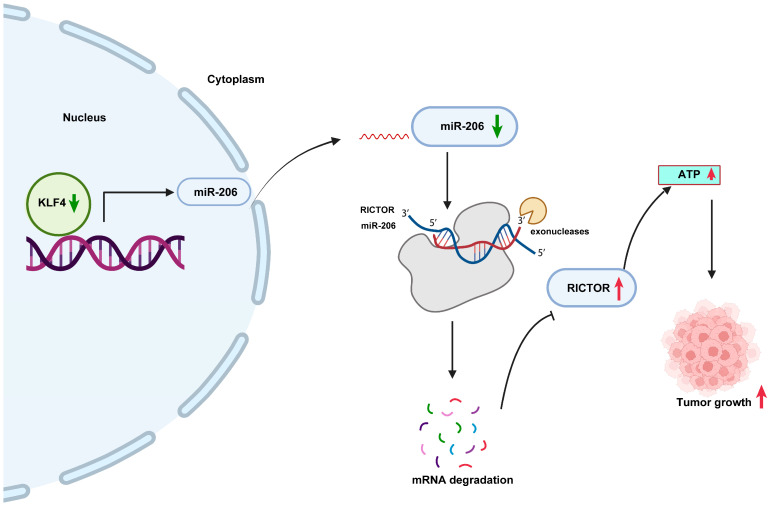
Schematic diagram illustrating the underlying mechanisms of KLF4 in suppressing the progression of hepatocellular carcinoma by reducing tumor ATP synthesis through targeting the miR-206/RICTOR axis. (Green arrows represent decrease, red arrows represent increase).

## Data Availability

All data are available in main text or Appendix A.

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
