# Peer review of "KLF4 Suppresses the Progression of Hepatocellular Carcinoma by Reducing Tumor ATP Synthesis through Targeting the Mir-206/RICTOR Axis"

_ijms, 2024, doi:10.3390/ijms25137165_

Round 1

Reviewer 1 Report

Comments and Suggestions for Authors

The current research manuscript “KLF4 suppresses the progression of hepatocellular carcinoma by reducing tumor ATP synthesis through targeting the miR-206/RICTOR axis” by Wang et al is focused on important function of KLF4 and impact of KLF 4 on tumor progression on miR206/RICTOR axis. Metabolic axis (OXPHOS) for tumor is an important aspect of study. I would like to commend the authors on their excellent elaborate work and subsequently drafted well-written descriptive manuscript with novel findings that help advance the hepatocellular carcinoma collective invasion field.

My comments and suggestions below should be seen as points that can hopefully add to this already well structure work, none of which should preclude eventual acceptance of this study.

Material and Method:

Visible scale bar in IF images

shKLF4 sequence (Ref)

Overall, this is an excellent study, detailed in a well-written manuscript with a clarity of figures that is commendable. There is substantial supplementary information provided, all of which supports the conclusions here, and the authors have done a really good job on ensuring that the narrative is clearly articulated and represented in the main figures.

Reviewer 2 Report

Comments and Suggestions for Authors

In this study, the authors attempt to investigate the role of KLF4 within hepatocellular carcinoma tumorigenesis. As a result, the study discovers that KLF4 can reduce ATP synthesis by suppressing RICTOR expression via regulating miR-206 expression in hepatocellular carcinoma model, suggesting the KLF4-miR206-RICTOR axis as potential therapeutic target in treating hepatocellular carcinoma. The study is well-designed and well-supported by its findings, hence can be accepted as its current state. I have no further comment for this study.
